# Identification of risk factors associated with prolonged hospital stay following primary knee replacement surgery: a retrospective, longitudinal observational study

Rebecca Wilson [1,2] Ruta Margelyte [3] Maria Theresa Redaniel,[1,2] Emily Eyles [1,2] Tim Jones [1,2] Chris Penfold [1,2] Ashley Blom,[3] Andrew Elliott,[4] Alison Harper,[5,6] Tim Keen,[4] Martin Pitt,[5,6] Andrew Judge[7]

For numbered affiliations see end of article.

**Correspondence to**
Dr Rebecca Wilson;
rebecca.wilson@bristol.ac.uk

## ABSTRACT

**Objectives** To identify risk factors associated with prolonged length of hospital stay and staying in hospital longer than medically necessary following primary knee replacement surgery.

**Design** Retrospective, longitudinal observational study.

**Setting** Elective knee replacement surgeries between 2016 and 2019 were identified using routinely collected data from an NHS Trust in England.

**Participants** There were 2295 knee replacement patients with complete data included in analysis. The mean age was 68 (SD 11) and 60% were female.

**Outcome measures** We assessed a binary length of stay outcome (>7 days), a continuous length of stay outcome (≤30 days) and a binary measure of whether patients remained in hospital when they were medically fit for discharge.

**Results** The mean length of stay was 5.0 days (SD 3.9), 15.4% of patients were in hospital for >7 days and 7.1% remained in hospital when they were medically fit for discharge. Longer length of stay was associated with older age (b=0.08, 95% CI 0.07 to 0.09), female sex (b=0.36, 95% CI 0.06 to 0.67), high deprivation (b=0.98, 95% CI 0.47 to 1.48) and more comorbidities (b=2.48, 95% CI 0.15 to 4.81). Remaining in hospital beyond being medically fit for discharge was associated with older age (OR=1.07, 95% CI 1.05 to 1.09), female sex (OR=1.71, 95% CI 1.19 to 2.47) and high deprivation (OR=2.27, 95% CI 1.27 to 4.06).

**Conclusions** The regression models could be used to identify which patients are likely to occupy hospital beds for longer. This could be helpful in scheduling operations to aid hospital efficiency by planning these patients' operations for when the hospital is less busy.

## STRENGTHS AND LIMITATIONS OF THIS STUDY

⇒ Robust statistical methods used to develop statistical models with good discrimination and calibration.
⇒ Consistent findings across three output measures of hospital throughput.
⇒ Single-centre study may restrict generalisability of findings.
⇒ Data do not include operations during or after the COVID-19 pandemic, after which the healthcare landscape changed and waiting list times have increased further.

## INTRODUCTION

Osteoarthritis is a common disease and a leading cause of disability worldwide,[1] with pain being the primary reason that people seek medical care. In those with end stage disease, joint replacement surgery is a well-established, common and highly effective surgical procedure,[2] where the majority of patients achieve substantial reductions in pain, improved joint function, mobility and health-related quality of life.[3] These operations cost the National Health Service (NHS) approximately £2 billion annually, which represents almost 1.5% of the entire NHS budget.[4] Total knee replacements are a common surgery in the UK, with up to 100 000 performed each year[5] and with a lifetime risk of 10.8% for women and 8.1% for men undergoing knee replacement.[6] The number of knee replacement surgeries is increasing in number due to an ageing and increasingly obese population[7 8] and are expected to continue to rise by 40% over the next 40 years.[9 10]

Waiting times for NHS care have increased in recent years,[11] while hospitals' capacity (the number of available theatres, surgeons and beds) is declining.[7] NHS acute hospital trusts face challenges in planning hospital bed capacity, especially in winter months when patient demand for hospital care is very high. In the UK, prior to the COVID-19 pandemic,

BMJ

substantial challenges existed for delivering timely joint replacement surgery, with orthopaedics having the largest waiting list of any individual surgical specialty.[12] Furthermore, planned operations are commonly cancelled, with 60% of cancellations happening on the day of surgery, usually due to lack of beds, staff and/or operating capacity, which can be compounded during the winter months.[13]

Patients remaining in hospital for longer than is necessary following surgery result in fewer hospital beds being available for the next scheduled patients. The time patients should expect to remain in hospital following total knee replacement surgery has decreased in recent years (from 3.7 days to 3.0 days between 2006 and 2016),[14] which suggests that hospital capacity and waiting lists could have scope to improve. Furthermore, shorter length of hospital stay does not reflect compromised patient care[15] and is financially more sustainable.

Waiting lists could be reduced if hospitals are able to plan the timing of surgeries in order to optimise hospital efficiency. There is a need to identify predictors of hospital throughput, to better plan elective surgery, by understanding the characteristics of patients at greatest risk of staying in hospital longer than medically necessary.

The aim of this study was to identify risk factors associated with prolonged length of hospital stay and staying in hospital longer than medically necessary following primary knee replacement surgery using linked routinely collected data from an NHS Trust in South-West England.

## METHODS
### Data sources
This is a longitudinal observational study and is reported in line with the STROBE reporting guidelines.[16] Routinely collected data from an NHS Trust in England were used to identify patients receiving elective primary total knee replacement surgeries between 2016 and 2019. The Trust's electronic health records (EHR) were used to identify elective total knee replacements, using a combination of OPCS4 procedure and surgical site codes (see online supplemental table S1). Tourniquet and wound drainage were not routinely used. Tranexamic acid was routinely administered intravenously intraoperatively. Further, the EHR were used to extract data for patient demographics (age, sex, deprivation quintile and comorbidities) and for admission details (length of stay and time and date of admission and discharge). Information describing the Trust's ratio of emergency to elective surgeries, daily emergency (non-elective) admissions and daily non-elective occupied beds were obtained from the Hospital Episode Statistics admitted patient care (HES-APC) dataset.[17]

### Outcome variables
A continuous measure of length of stay was generated to summarise the number of days between admission and discharge, up to 30 days; those in hospital for more than 30 days were coded as missing for this variable (this was to exclude outliers and excluded only 0.92% (N=21) of complete cases from analysis). Two binary outcome variables were derived, one indicating if patient's hospital stay was longer than 7 days (0= ≤7 days, 1= >7 days) and one indicating if patient had a recorded medically fit for discharge date that preceded their discharge date (0=did not have a medically fit for discharge date recorded, 1=patient remained in hospital when they were medically fit for discharge). Patients without a medically fit for discharge date were assumed to have been discharged when medically fit.

### Predictive factors
Variables in the multivariable models include: a continuous measure of age on admission, patient sex (0=male, 1=female), patient area-level deprivation (Index of Multiple Deprivation (IMD) quintile derived from lower super output areas (LSOA)[18]), comorbidities (as measured by the weighted Charlson Comorbidity Index[19]), a categorical measure of time since last discharge (0–2 months, 2–12 months, 12 months or more, or no previous admissions), a categorical measure of admission hour (06.00–12.00, 12.00–18.00, 18.00–06.00), the day of the week, season (winter: December–February; spring: March–May; summer: June–August; autumn: September–November) and year of admission, and the Trust monthly emergency/elective admissions ratio, daily non-elective admissions, and daily non-elective occupied beds.

### Statistical analysis
Sample characteristics were summarised for all patients. To explore bias in data availability, descriptive statistics for key patient characteristics (age, sex, socioeconomic deprivation, ethnicity and comorbidities) were summarised for three groups: all patients, those with complete case data and those who had a previous discharge. No notable differences in these characteristics were observed between these subsamples (see online supplemental table S2). Complete case analysis was used throughout. The complete case sample was n=2295, or 76.3% of the whole sample.

Univariable and multivariable regression models were used to test associations between predictive factors and outcomes. Model assumptions were checked, and the following interactions were tested for: age and sex, age and comorbidities, age and season, age and deprivation, season and comorbidities, deprivation and comorbidities, season and deprivation. No interactions were observed. There was some evidence to suggest the continuous outcome (length of stay) violated the assumptions for the linear regression model (residuals not normally distributed), so robust standard errors were estimated using the Huber-White sandwich estimator.[20]

Multivariable logistic regression models were used for binary outcomes and linear regression for the continuous outcome. The same 12 potential predictors (age, sex, area-level deprivation, comorbidities, days since last discharge,

elective/non-elective ratio, non-elective admissions, non-elective occupied beds, admission hour and year, day and season of admission) were entered into the models for all outcomes. For each outcome, we ran least absolute shrinkage and selection operator (LASSO), elastic net and ridge regression models. To produce the final models, we performed multivariable logistic (for binary outcomes)/linear (for continuous outcome) regression using backwards variable selection at p=0.1 and obtained C-statistics for the binary outcomes and $R^2$ for the continuous outcome. Calibration plots were produced for the final models for the binary outcomes.

Using the predictor variables retained in the backwards selection regression models, forest plots were produced to display the effects of each predictive factor for each outcome. All analyses were performed using Stata V.17.[21]

### Patient and public involvement

Patients and the public were consulted in a workshop for suggestions and comments to inform the development of the grant. They were again consulted in a follow-up workshop where initial results were presented. They were asked about their opinion on the prioritisation of particular patient groups (eg, less complex patients during winter and complex patients in the summer) to maximise efficiency for surgery waiting lists.

### Ethics approval

We were provided with routinely collected HES data under licence from NHS Digital (DARS-NIC-17875-X7K1V). The licence allows us to use the information under Section 261 of the Health and Social Care Act 2012, 2(b)(ii): 'after taking into account the public interest as well as the interests of the relevant person, considers that it is appropriate for the information to be disseminated'.

### Role of the funding source

The funders had no role in any of the following: the study design; the collection, analysis, and interpretation of data; the writing of the report; the decision to submit the paper for publication.

### RESULTS

Between January 2016 and December 2019, 3008 primary elective knee replacements were performed. Of these, 2295 (76.3%) patients had data available for all variables and were included in the complete case analysis (see figure 1). As the continuous length of stay outcome measure was capped at 30 days, the multivariable analyses for this outcome included 2274 patients. Descriptive statistics are presented in table 1 for the complete case sample. The mean length of stay for patients who were in hospital for 30 days or less was 5.0 days (SD=3.9), 15.4% of patients stayed in hospital for more than 7 days. Most patients (92.9%) did not remain in hospital once they were medically fit for discharge, but 7.1% did have a

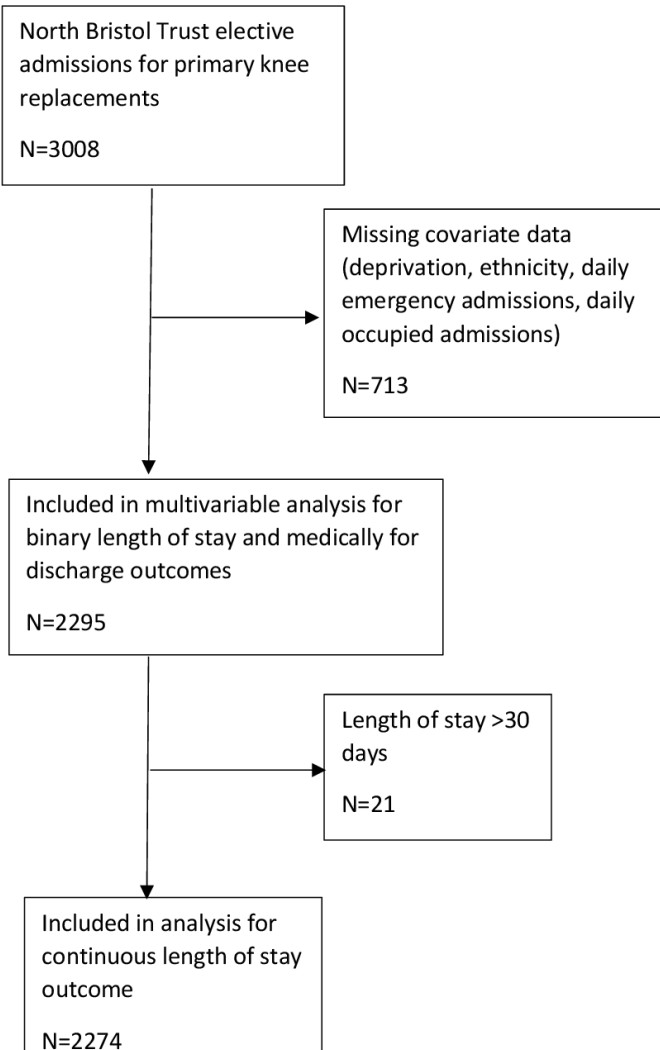

**Figure 1** Flow diagram of patients excluded and included in the study.

recorded medically fit for discharge date which preceded their discharge date.

Multivariable backwards regression models are presented in table 2 and in figures 2–4. The full multivariable regression models, LASSO, elastic net and ridge regression models for each outcome are presented in online supplemental tables S3–S5. The models for the binary outcomes (length of stay and medically fit for discharge) are accompanied by calibration plots (online supplemental figures S1–S4).

Remaining in hospital for more than 7 days following surgery was associated with older age, higher deprivation, more comorbidities and being recently discharged from hospital (see table 2/figure 2). Associations between length of stay and time of admission were also observed but the CIs were wide. Year and day of admission were also associated with length of the stay but without any apparent trend. No associations were found for sex, season of admission or hospital-related factors (emergency over elective admissions ratio, non-elective admissions ratio or non-elective occupied beds).

**Table 1** Characteristics of complete cases who have undergone North Bristol Trust elective admissions for primary knee replacements (2015–2019) (N=2295)

| Variable | N (%)* | Proportion (%) admitted >7 days | Mean length of stay (SD) N=2274 | Proportion (%) medically fit for discharge |
|---|---|---|---|---|
| Age at admission | | | | |
| Age at admission, mean (SD) | 67.4 (11.3) | – | – | – |
| Sex | | | | |
| Female | 1381 (60.2) | 15.3 | 5.1 (3.9) (N=1367) | 8.5 |
| Male | 914 (39.8) | 15.5 | 4.7 (3.8) (N=907) | 5.0 |
| Deprivation (Index Multiple Deprivation score) | | | | |
| 1 (least deprived) | 588 (25.6) | 11.1 | 4.4 (3.1) (N=583) | |
| 2 | 564 (24.6) | 18.3 | 5.1 (4.0) (N=555) | |
| 3 | 395 (17.2) | 17.0 | 5.2 (4.5) (N=391) | |
| 4 | 418 (18.2) | 14.8 | 5.0 (3.7) (N=416) | |
| 5 (most deprived) | 330 (14.4) | 17.0 | 5.2 (4.2) (N=329) | |
| Ethnicity† | | | | |
| Non-white | 76 (5.3) | – | – | – |
| Asian | 25 (1.7) | – | – | – |
| Black | 31 (2.2) | – | – | – |
| Mixed | 10 (0.7) | – | – | – |
| Other | 10 (0.7) | – | – | – |
| White | 1368 (94.7) | – | – | – |
| Unknown | 851 | – | – | – |
| Comorbidities (weighted Charlson index) | | | | |
| 0 | 1260 (54.9) | 12.0 | 4.5 (3.4) (N=1255) | 4.6 |
| 1–2 | 867 (37.8) | 17.9 | 5.4 (4.1) (N=854) | 9.5 |
| 3–4 | 148 (6.5) | 26.4 | 6.6 (5.3) (N=145) | 15.5 |
| ≥5 | 20 (0.9) | 8.0 | 7.8 (5.5) | 0 |
| Time since last discharge | | | | |
| 0–2 months | 338 (14.7) | 26.9 | 6.3 (5.0) (N=332) | |
| 2–12 months | 474 (20.7) | 18.8 | 5.5 (4.2) (N=470) | |
| 12 months or more | 409 (17.8) | 13.0 | 4.6 (3.4) | |
| Never | 1074 (46.8) | 11.1 | 4.4 (3.3) (N=1063) | |
| Emergency over elective admissions ratio (general/acute, monthly), mean±SD | 0.8±0.1 | – | – | – |
| Emergency admissions, daily, mean±SD | 152.6±21.2 | – | – | – |
| Emergency occupied beds, daily, mean±SD | 914.9±43.4 | – | – | – |
| Admission hour category | | | | |
| 24.00–06.00 | 2 (0.1) | 50.0 | 9.5 (3.5) | 0 |
| 06.00–12.00 | 2092 (91.2) | 14.4 | 4.9 (3.7) (N=2075) | 6.7 |
| 12.00–18.00 | 195 (8.5) | 22.6 | 5.6 (4.7) (N=192) | 9.7 |
| 18.00–24.00 | 6 (0.3) | 100.0 | 14.2 (7.0) (N=5) | 50.0 |
| Year of admission | | | | |
| 2016 | 185 (8.1) | 10.8 | 4.7 (3.1) | 2.7 |
| 2017 | 783 (34.1) | 15.6 | 4.9 (3.6) (N=770) | 7.2 |
| 2018 | 629 (27.4) | 18.4 | 5.3 (4.1) (N=626) | 9.1 |
| 2019 | 698 (30.4) | 13.6 | 4.8 (4.1) (N=693) | 6.5 |
| Month of admission | | | | |
| January | 131 (5.7) | 20.6 | 5.4 (4.7) | 8.4 |

**Table 1**  Continued

| Variable | N (%)* | Proportion (%) admitted >7 days | Mean length of stay (SD) N=2274 | Proportion (%) medically fit for discharge |
|---|---|---|---|---|
| February | 170 (7.4) | 14.1 | 4.9 (4.3) | 3.5 |
| March | 208 (9.1) | 18.8 | 5.1 (4.2) (N=204) | 7.2 |
| April | 183 (8.0) | 17.5 | 4.9 (3.0) (N=181) | 6.6 |
| May | 194 (8.5) | 17.0 | 5.1 (3.8) (N=192) | 9.3 |
| June | 194 (8.5) | 9.8 | 4.4 (3.0) (N=192) | 7.2 |
| July | 188 (8.2) | 16.0 | 4.9 (4.1) (N=185) | 10.6 |
| August | 164 (7.2) | 15.2 | 4.9 (3.9) (N=161) | 5.5 |
| September | 187 (8.2) | 14.4 | 5.2 (4.1) | 8.0 |
| October | 254 (11.1) | 15.8 | 5.0 (4.0) (N=251) | 5.9 |
| November | 260 (11.3) | 13.5 | 4.9 (3.8) (N=258) | 8.1 |
| December | 162 (7.1) | 13.6 | 5.0 (3.5) | 4.3 |
| Day of the week of admission | | | | |
| Sunday | 2 (0.1) | 50.0 | 8.5 (10.6) | 50.0 |
| Monday | 495 (21.6) | 18.4 | 4.8 (3.8) (N=490) | 5.5 |
| Tuesday | 467 (20.4) | 17.3 | 4.9 (3.7) (N=461) | 8.8 |
| Wednesday | 403 (17.6) | 13.2 | 5.0 (4.4) (N=399) | 7.7 |
| Thursday | 408 (17.8) | 14.2 | 5.0 (3.5) (N=406) | 6.1 |
| Friday | 467 (20.4) | 13.9 | 5.1 (3.9) (N=463) | 7.5 |
| Saturday | 53 (2.3) | 7.6 | 4.7 (3.2) (N=53) | 5.7 |
| Season of admission | | | | |
| Winter (Dec–Feb) | 463 (20.2) | 15.8 | 5.1 (4.1) | 5.2 |
| Spring (Mar–May) | 585 (25.5) | 17.8 | 5.0 (3.7) (N=577) | 7.7 |
| Summer (Jun–Aug) | 546 (23.8) | 13.6 | 4.7 (3.7) (N=538) | 7.9 |
| Autumn (Sep–Nov) | 701 (30.5) | 14.6 | 5.0 (3.9) (N=696) | 7.3 |
| Spell length of stay, mean±SD | 5.5±6.1 | – | – | – |
| Spell length of stay up to 30 days, mean±SD, (N=2274) | 5.0±3.9 | – | – | – |
| Patients staying >7 days | | | | |
| ≤7 (no) | 1942 (84.6) | – | – | – |
| >7 (yes) | 353 (15.4) | – | – | – |
| Patients medically fit for discharge (MFFD) | | | | |
| With MFFD date before discharge | 163 (7.1) | – | – | – |
| No MFFD date | 2132 (92.9) | – | – | – |
| Days between MFFD date and discharge, mean±SD, (N=163) | 7.2±13.7 | – | – | – |

*n (%) for categorical, mean±SD for continuous variables.
†Percentages are for known data. Not included in analysis due to large number of 'unknowns'.
IMD, Index Multiple Deprivation; LOS, length of stay; NBT, North Bristol Trust.

Similarly, longer length of stay following surgery (measured using a continuous variable ranging from 0 to 30 days) was associated with older age, increasing deprivation, more comorbidities and recently being discharged from hospital (compared with never having been admitted to hospital or being discharged from hospital at least over 2 months ago). Female sex was also associated with the continuous measure of length of stay (see table 2/figure 3). A spurious association with a wide CI was again observed for admission hour. No associations were observed for hospital-related factors, admission year, day or season.

Remaining in hospital when being medically fit for discharge was associated with older age, being female, greater deprivation, more comorbidities (though not the highest category of comorbidities, which was dropped from the model), being discharged from hospital recently (compared with 12 months or more or never), surgeries

**Table 2** Predictors of length of stay and remaining in hospital beyond being medically fit for discharge

| Variable | Length of stay (admissions>7 days) N=2295 | Length of stay (≤ 30 days) N=22 74* | Medically fit for discharge N=22 75† |
|---|---|---|---|
| | OR (95% CI) | Coef (95% CI) | OR (95% CI) |
| Age at admission | 1.06 (1.04 to 1.07)* | 0.08 (0.07 to 0.09)* | 1.07 (1.05 to 1.09)* |
| Sex (female vs male) | – | 0.36 (0.06 to 0.67)* | 1.71 (1.19 to 2.47)* |
| Deprivation (Inedx Multiple Deprivation score) | | | |
| 1 (least deprived) | 1.00 | 0.00 | 1.00 |
| 2 | 1.74 (1.22 to 2.47)* | 0.65 (0.26 to 1.04)* | 1.76 (1.05 to 2.96)* |
| 3 | 1.74 (1.18 to 2.58)* | 0.80 (0.32 to 1.28)* | 1.84 (1.05 to 3.23)* |
| 4 | 1.54 (1.04 to 2.28)* | 0.75 (0.33 to 1.16)* | 2.07 (1.20 to 3.57)* |
| 5 (most deprived) | 2.03 (1.34 to 3.05)* | 0.98 (0.47 to 1.48)* | 2.27 (1.27 to 4.06)* |
| Comorbidities (weighted Charlson Comorbidities index) | | | |
| 0 | 1.00 | 0.00 | 1.00 |
| 1–2 | 1.40 (1.08 to 1.81)* | 0.63 (0.31 to 0.94)* | 1.89 (1.32 to 2.71)* |
| 3–4 | 1.85 (1.20 to 2.86)* | 1.42 (0.55 to 2.29)* | 2.56 (1.48 to 4.42)* |
| ≥5 | 3.03 (1.14 to 8.05)* | 2.48 (0.15 to 4.81)* | – |
| Time since last discharge | | | |
| 0–2 months | 1.00 | 0.00 | 1.00 |
| 2–12 months | 0.67 (0.47 to 0.95)* | −0.65 (−1.29 to −0.02)* | 0.97 (0.60 to 1.57) |
| 12 months or more | 0.41 (0.27 to 0.61)* | −1.45 (−2.06 to −0.84)* | 0.53 (0.30 to 0.92)* |
| Never | 0.38 (0.28 to 0.53)* | −1.49 (−2.05 to −0.93)* | 0.53 (0.33 to 0.84)* |
| Emergency over elective admissions ratio | – | – | – |
| Non-Elective admissions | – | – | – |
| Non-Elective occupied beds | – | – | – |
| Admission hour category | | | |
| 06.00–12.00 | 1.00 | 0.00 | 1.00 |
| 12.00–18.00 | 1.81 (1.23 to 2.66)* | 0.79 (0.17 to 1.41)* | 1.56 (0.91 to 2.66) |
| 18.00-06:00 | 77.21 (8.01 to 744.20)* | 7.55 (2.85 to 12.24)* | 8.11 (1.74 to 37.75)* |
| Year of admission | | | |
| 2016 | 1.00 | 0.00 | 1.00 |
| 2017 | 1.66 (0.96 to 2.86) | 0.20 (−0.27 to 0.67) | 2.95 (1.14 to 7.64)* |
| 2018 | 1.96 (1.12 to 3.42)* | 0.47 (−0.04 to 0.99) | 3.89 (1.49 to 10.12)* |
| 2019 | 1.14 (0.65 to 1.99) | −0.21 (−0.71 to 0.30) | 2.32 (0.88 to 6.09) |
| Day of admission | | | |
| Sunday | – | – | – |
| Monday | 1.00 | – | – |
| Tuesday | 0.89 (0.62 to 1.26) | – | – |
| Wednesday | 0.59 (0.40 to 0.88)* | – | – |
| Thursday | 0.66 (0.45 to 0.97)* | – | – |
| Friday | 0.78 (0.54 to 1.13) | – | – |
| Saturday | 0.20 (0.06 to 0.73)* | – | – |
| Season of admission | | | |
| Winter (Dec–Feb) | 1.00 | – | – |
| Spring (Mar–May) | 1.14 (0.80 to 1.61) | – | – |
| Summer (Jun–Aug) | 0.73 (0.50 to 1.06) | – | – |
| Autumn (Sep–Nov) | 0.90 (0.64 to 1.28) | – | – |
| | C-statistic (95% CI) | R² | C-statistic (95% CI) |
| | 0.72 (0.69 to 0.74) | 0.130 | 0.74 (0.71 to 0.78) |

**Table 2** Continued

| Variable | Length of stay (admissions>7 days) N=2295 | Length of stay (≤ 30 days) N=22 74* | Medically fit for discharge N=22 75† |
|---|---|---|---|
| | OR (95% CI) | Coef (95% CI) | OR (95% CI) |

*included sample N=2274 patients with length of stay ≤30 days.
†included sample N=2295 minus those excluded when CCI score ≥5 was dropped from the model (N=20).

in 2017 and 2018 (compared with 2016). A weak association was observed with admission hour and no associations were found for hospital-related factors, admission day or season (table 2/figure 4).

The calibration plots (figures 5 and 6) indicate close agreement for predicted and observed values for the binary length of stay and the medically fit for discharge outcome. The respective C-statistics of 0.72 and 0.74 also indicated good model discrimination.

## DISCUSSION
### Summary of main findings
In this study, we observed that over 7% of patients remained in hospital beyond the date they were medically fit to be discharged, and 15% had long stays in hospital of more than 7 days. Being able to reduce this burden on hospital throughput would have a positive impact on overall hospital efficiency. Risk factors associated with remaining in hospital longer, and for longer than is necessary, included older age, female patients with more comorbidities and those recently admitted to hospital.

Patients from more deprived areas also had worse outcomes. There was no evidence of interactions with season, suggesting that efforts to limit operations to older patients and those with co-morbidities to summer months will not impact on hospital efficiency and throughput.

### What is already known
Our results were in line with previous research showing that longer length of stay is associated with older age, being female and worse health.[14 22–24] Hospital and surgeon-related factors (including hospital site, surgeon, time and day of surgery)[22] were also cited as predictors of length of stay, some of which are modifiable and presents opportunities to optimise efficiency, however, we found that hospital-related factors were not predictors of length of stay, although variables in this study, such as admission hour and day, were limited by small numbers in some categories. The factors associated with outcomes in our study were all patientrelated, including health-related and sociodemographic.

### Strengths and limitations
This observational study had many strengths, including its reasonable sample size and use of robust statistical methods. This allowed us to identify a number of predictors with good discrimination and calibration. We also utilised three metrics of throughput as outcomes which demonstrated consistent findings. There are of course limitations to our study. There is great variation in

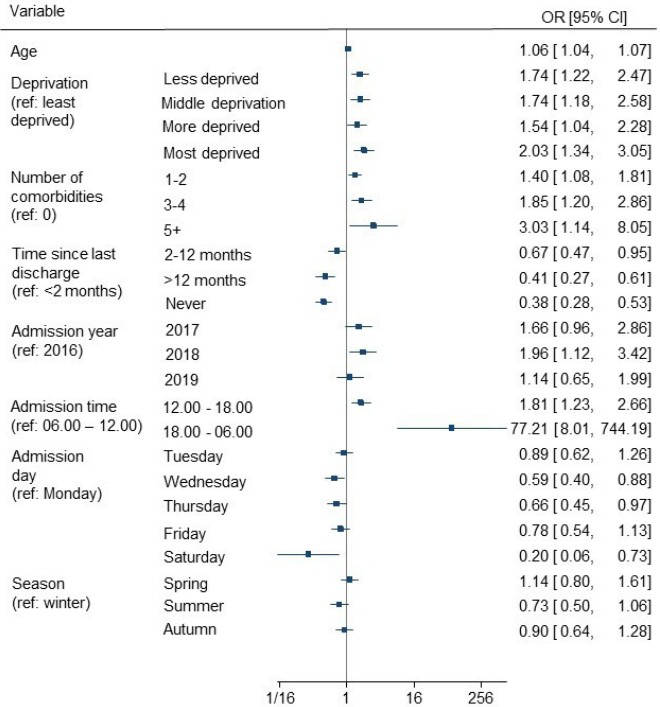

**Figure 2** Predictors of length of stay of more than 7 days for patients following knee replacement surgery.

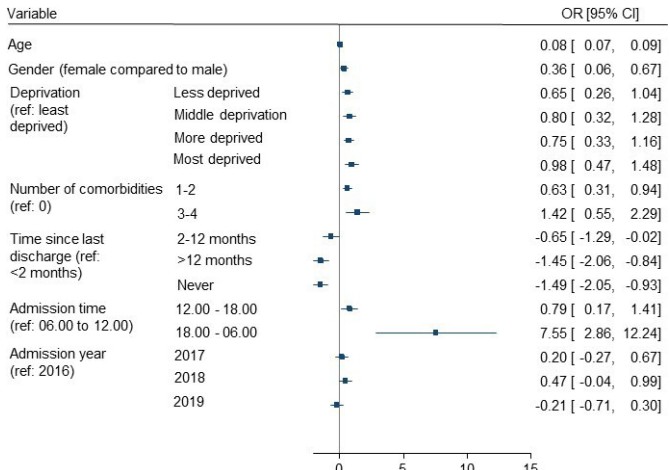

**Figure 3** Predictors of length of stay (using a continuous measure) in patients following knee replacement surgery.

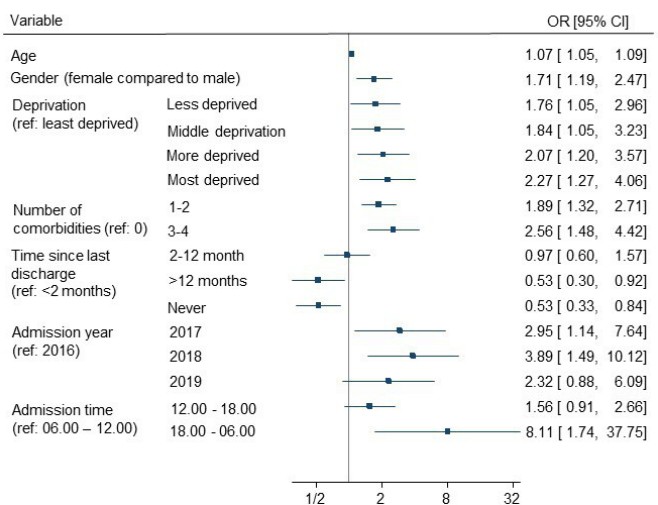

Figure 4 Predictors of remaining in hospital beyond being declared medically fit following knee replacement surgery.

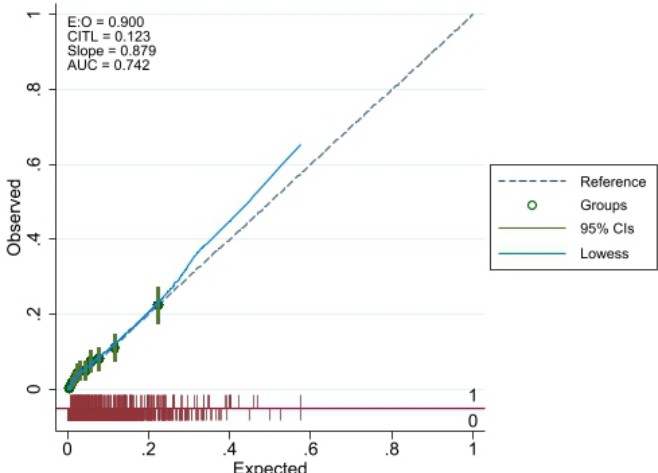

Figure 6 Calibration plot for final model for medically fit for discharge (E:O, Expected:Observed; CITL, Calibration in the large; AUC, Area under the curve) outcome.

practice and provision of services across the country[7] and, as a single-centre study, the results may not be generalisable to other NHS trusts and settings. We only included data on operations performed in the NHS Trust and not privately funded patients. We purposefully analysed data from before the start of the COVID-19 pandemic, after which the healthcare landscape changed extraordinarily. Waiting times for knee replacement surgery have since increased, leading to huge burden on patients' well-being,[25] and this is not represented in our results. There were missing data, however, comparison of patient characteristics of those in full vs complete case dataset shows no evidence of selection (responder) bias (see online supplemental table S2).

### Implications for practice

Our models would allow healthcare providers to identify which patients are more likely to remain in hospital for

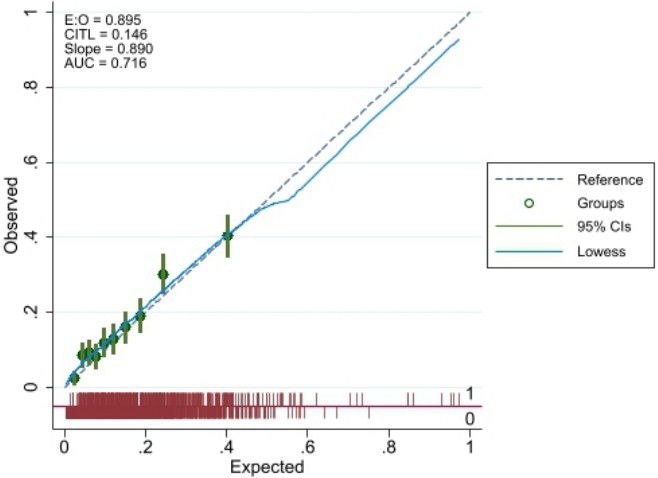

Figure 5 Calibration plot for final model for binary length of stay outcome (E:O, Expected:Observed; CITL, Calibration in the large; AUC, Area under the curve).

longer, using these known variables, or risk factors. While providers would not be able to change these risk factors, this knowledge could be used in planning surgeries and hospital capacity, potentially allowing the more complex surgeries with a longer predicted in-patient stay to be scheduled for when the hospital is under less strain, for example, in the summer months.

Rescheduling patients' surgeries due to personal characteristics could be considered controversial, or potentially introducing inequalities. However, when asked if they would mind if surgeries were scheduled to a particular time based on patient characteristics, our PPI group said not, providing they were kept informed and could rely on the information being told to them. If there is evidence to support rescheduling patients in order to improve efficiency, patients and their caregivers could accept surgeries being moved.

We were able to build good-fitting models of length of stay and staying in hospital beyond being medically fit for discharge using data from all total knee replacements between 2016 and 2019 which can identify which patients are more likely to stay in hospital for longer and for longer than is necessary. These models may be used in planning bed capacity and potentially reduce waiting list times.

**Author affiliations**
[1]The National Institute for Health Research Applied Research Collaboration West (NIHR ARC West), University Hospitals Bristol and Weston NHS Foundation Trust, Bristol, UK
[2]Population Health Sciences, Bristol Medical School, University of Bristol, Bristol, UK
[3]Translational Health Sciences, Bristol Medical School, University of Bristol, Bristol, UK
[4]North Bristol NHS Trust, Bristol, UK
[5]The National Institute for Health Research Applied Research Collaboration South-West Peninsula (PenARC), University of Exeter, Exeter, UK
[6]Medical School, University of Exeter, Exeter, UK
[7]Translational Health Sciences, University of Bristol, Bristol, UK

**Contributors** RW, RM, MTR, and AJ drafted the original manuscript. RW and RM analysed the data. RW, RM, MTR, EE, TJ, CP, AWB, AE, AH, TK, MP and AJ all contributed to the conception and planning of this project and all reviewed and revised the manuscript. MTR and AJ supervised the project. MTR is the guarantor for the project.

**Funding** This study was funded by the Health Data Research (HDR) UK South West Better Care Partnership (#6.12). This research was supported by the National Institute for Health Research (NIHR) Applied Research Collaboration West (ARC West) at University Hospitals Bristol and Weston NHS Foundation Trust (core NIHR infrastructure funded: NIHR200181).

**Competing interests** None declared.

**Patient and public involvement** Patients and/or the public were involved in the design, or conduct, or reporting, or dissemination plans of this research. Refer to the Methods section for further details.

**Patient consent for publication** Not applicable.

**Ethics approval** We were provided with pseudonymised North Bristol Trust hospital admissions data under the NIHR ARC West Partnership Agreement. The project received ethical approval from the University of Bristol Faculty of Health Sciences ethical review board on 3 November 2020 (ref# 109024).

**Provenance and peer review** Not commissioned; externally peer reviewed.

**Data availability statement** Data may be obtained from a third party and are not publicly available. The data used in the study are collected by the North Bristol Trust (NBT) as part of their care and support. Sharing of anonymised data with the University of Bristol was underpinned by a data sharing agreement and solely covers the purposes of this study. Data requests can be made through the NBT.

**ORCID iDs**
Rebecca Wilson http://orcid.org/0000-0003-4709-7260
Ruta Margelyte http://orcid.org/0000-0002-7914-8037
Emily Eyles http://orcid.org/0000-0002-2695-7172
Tim Jones http://orcid.org/0000-0002-1199-8668
Chris Penfold http://orcid.org/0000-0001-8654-353X

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
