## [Reviewer comments · BMJ Open]

ARTICLE DETAILS

TITLE (PROVISIONAL)	Identification of risk factors associated with prolonged hospital stay following primary knee replacement surgery: a retrospective, longitudinal observational study
AUTHORS	wilson, rebecca; Margelyte, Ruta; Redaniel, Maria Theresa; Eyles, Emily; Jones, Tim; Penfold, Chris; Blom, AW; Elliott, Andrew; Harper, Alison; Keen, Tim; Pitt, Martin; Judge, Andrew

VERSION 1 – REVIEW

REVIEWER	Samo Fokter University Clinical Centre, Maribor, Slovenia
REVIEW RETURNED	18-Oct-2022

GENERAL COMMENTS	I commend the authors for performing their research entitled "Identification of risk factors associated with prolonged hospital stay following primary knee replacement surgery: a longitudinal observational study". I found the topic very interesting, the manuscript well written, the methods sound, and the conclusions based on the results. I would only suggest to include some basic clinical data in the manuscript: (1) did the health care provider follow the "fast track surgery" protocol; (2) what was the proportion of total knee replacement versus partial knee replacement in the cohort; and (3) which surgical principles were in routine use during the observational period (tourniquet, wound drainage, cyklokapron (tranexamic acid) administration protocol, etc.).
--

REVIEWER	Angelina Müller Goethe-Universität Frankfurt am Main
REVIEW RETURNED	10-Nov-2022

GENERAL COMMENTS	This study investigates risk factors associated with longer hospital stays than medically necessary after primary knee replacement surgery using routine data. It is well and logically explained why this research can improve planning hospital stays in advance - something that could have great positive impact on future organization of processes in hospital. Introduction: Very well written making it easy to understand the main goals of the study and why this research is necessary - also well comprehensible for readers not familiar with processes in UK hospitals. Methods: Robust statistical models in line with STROBE and well chosen for routine data analysis. Results: Very well presented and easy to follow especially by showing
---

	directly what it was compared to. All tables well structured. Discussion: Strengths and limitations well discribed. Appropriate suggestions for implementation of study results. Ethics approval is missing!
--	---

VERSION 1 – AUTHOR RESPONSE

In response to Reviewer 1:

Thank you for your generous comments and for highlighting some of the clinical oversights in our submission. In response to your comments:

1. "did the health care provider follow the "fast track surgery" protocol?"

Thank you for questioning this, unfortunately the reviewers don't reference the "fast track surgery" protocol to which they are referring so we are unable to address this question. If the reviewers would like to specify which protocol they are referring to, we would be happy to confirm.

2. "what was the proportion of total knee replacement versus partial knee replacement in the cohort".

The surgeries are all total knee replacements, this was one of our inclusion criteria for surgeries. I realise that this is missing from the methods, so have included this in the Methods (page 6). Thank you for spotting this oversight.

3. "which surgical principles were in routine use during the observational period (tourniquet, wound drainage, cyklokapron (tranexamic acid) administration protocol?"

Thank for you this interesting question. Tourniquet and wound drainage were not routinely used. Tranexamic acid was routinely administered intravenously intraoperatively. I have added this information to the Methods (page 6).

In response to Reviewer 2, thank you for your kind comments, we really appreciate them and are glad you enjoyed the manuscript. We have added an Ethical Approval section to the Methods (page 8/9).

VERSION 2 – REVIEW

REVIEWER	Samo Fokter University Clinical Centre, Maribor, Slovenia
REVIEW RETURNED	17-Nov-2022
GENERAL COMMENTS	Thank you for your efforts. Your manuscript has been improved.